# ML$^2$-GCL: Manifold Learning Inspired Lightweight Graph Contrastive Learning

**Jianqing Liang** [1]  **Zhiqiang Li** [1]  **Xinkai Wei** [1]  **Yuan Liu** [1]  **Zhiqiang Wang** [1]

## Abstract

Graph contrastive learning has attracted great interest as a dominant and promising self-supervised representation learning approach in recent years. While existing works follow the basic principle of pulling positive pairs closer and pushing negative pairs far away, they still suffer from several critical problems, such as the underlying semantic disturbance brought by augmentation strategies, the failure of GCN in capturing long-range dependence, rigidness and inefficiency of node sampling techniques. To address these issues, we propose Manifold Learning Inspired Lightweight Graph Contrastive Learning (ML$^2$-GCL), which inherits the merits of both manifold learning and GCN. ML$^2$-GCL avoids the potential risks of semantic disturbance with only one single view. It achieves global nonlinear structure recovery from locally linear fits, which can make up for the defects of GCN. The most amazing advantage is about the lightweight due to its closed-form solution of positive pairs weights and removal of pairwise distances calculation. Theoretical analysis proves the existence of the optimal closed-form solution. Extensive empirical results on various benchmarks and evaluation protocols demonstrate effectiveness and lightweight of ML$^2$-GCL. We release the code at https://github.com/a-hou/ML2-GCL.

## 1. Introduction

Graph representation learning, which aims to learn appropriate low-dimensional representations for graph structured data to facilitate various downstream tasks (Cai et al., 2018), has shown wide applications in social network analysis (Pal et al., 2020), molecular property prediction (Liang et al., 2024) and point clouds (Du et al., 2024). Unlike traditional Euclidean data, unstructured features and complex relations in graphs pose unique challenges to representation learning. While a tremendous number of works have been developed (Kipf & Welling, 2017; Velickovic et al., 2018), the requirement of task-dependent labels limits their applicability in practical scenarios. Inspired by the success of contrastive learning in the visual domain, graph contrastive learning (GCL) has demonstrated great potential and yielded promising performance (Velickovic et al., 2019; Peng et al., 2020; Hassani & Khasahmadi, 2020; Zhu et al., 2020).

GCL pulls positive pairs closer and pushes negative pairs far away without labels to learn node embeddings for downstream tasks. Despite comparable performance to supervised learning methods, some critical challenges of augmentation strategies and node sampling approaches still remain to be solved. The commonly used random augmentation strategies such as node dropping (You et al., 2020) and edge perturbation (Qiu et al., 2020; Zhang et al., 2023), may destroy the structural integrity and semantic consistency of graphs. Current node sampling approaches mainly rely on the similarity of structure relations and node embeddings (Lee et al., 2022; Shen et al., 2023; Liang et al., 2025), or probability models, especially clustering algorithms or mixed models, to estimate the probability of positive pairs and negative pairs (Li et al., 2023; Dong et al., 2024; Sun et al., 2024). Such a simple and straightforward strategy may lead to high computational cost, and thus is not applicable to large-scale applications. Moreover, they also ignore the importance of different nodes, among which there exist significant differences in social networks. We are highly motivated to develop an efficient and effective GCL method significantly differs from existing approaches.

Manifold learning, which can capture both the local structure and global topology through modeling the geometric characteristics of data in low dimensional manifold space, provides a new perspective for graph contrastive learning. However, existing GCL methods have not fully exploited the potential of manifold learning yet, especially the lack

---

*Equal contribution [1]Key Laboratory of Computational Intelligence and Chinese Information Processing of Ministry of Education, School of Computer and Information Technology, Shanxi University, Taiyuan 030006, Shanxi, China. Correspondence to: Jianqing Liang <liangjq@sxu.edu.cn>.

*Proceedings of the 42$^{nd}$ International Conference on Machine Learning*, Vancouver, Canada. PMLR 267, 2025. Copyright 2025 by the author(s).

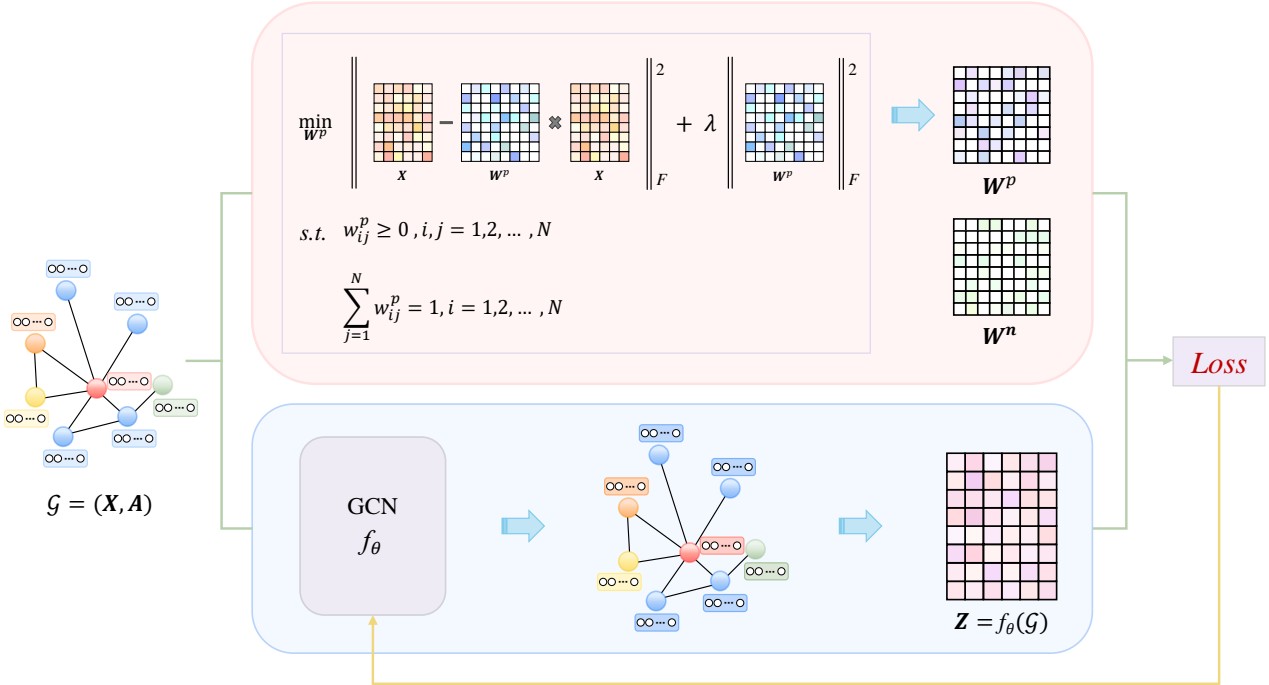

Figure 1. The overview of ML$^2$-GCL. With the original node feature $\boldsymbol{X}$ and the adjacency matrix $\boldsymbol{A}$, we compute the positive pairs weight matrix $\boldsymbol{W}^p$ with a closed-form solution of a manifold learning inspired node representation optimization problem. Then, we calculate the negative pairs weight matrix $\boldsymbol{W}^n$ and apply GCN as a graph encoder and obtain the node embedding $\boldsymbol{Z}$. Last, we compute the normalized loss function with regard to the positive pairs weight matrix $\boldsymbol{W}^p$, negative pairs weight matrix $\boldsymbol{W}^n$ as well as the node embedding $\boldsymbol{Z}$ and then update the graph encoder.

of theoretical guidance in node sampling. Inspired by LLE (Roweis & Saul, 2000), coherent structure of graphs leads to strong correlations between inputs, such as between neighboring nodes, generating observations that lie on or close to a smooth low-dimensional manifold.

In this paper, we propose Manifold Learning Inspired Lightweight Graph Contrastive Learning (ML$^2$-GCL). Figure 1 gives the overview of ML$^2$-GCL. Figure 2 shows the comparison of GRACE (Zhu et al., 2020), NCLA (Shen et al., 2023) and ML$^2$-GCL. Different from previous GCL methods, ML$^2$-GCL eliminates the need to estimate pairwise distances between node embeddings. It does not need to generate augmented views, thereby avoiding the underlying semantic disturbance. In addition, ML$^2$-GCL recovers global nonlinear structure from locally linear fits, which can better distinguish the importance of neighboring nodes. Our contributions are listed as follows.

- To the best of our knowledge, it is the first exploration to marry manifold learning with graph contrastive learning, which inherits the merits of both manifold learning and GCN.

- We propose ML$^2$-GCL solely relying on the original

graph with GCN to maintain semantic consistency and achieve lightweight.

- We design a novel contrastive loss function with the closed-form solution of anchor node reconstruction combination weights, which can better distinguish the importance of different nodes.

## 2. Related Work

### 2.1. Graph Contrastive Learning

Current GCL methods can be categorized into three mainstream paradigms, i.e., DGI (Velickovic et al., 2019), InfoNCE (Oord et al., 2018) and BGRL framework (Thakoor et al., 2022). DGI (Velickovic et al., 2019) aggregates the features of all nodes in graphs to obtain a global feature, and then maximizes the mutual information between the global feature and the node features. Based on this framework, MVGRL (Hassani & Khasahmadi, 2020) adopts graph diffusion and subgraph sampling strategies to advance the development of unsupervised graph contrastive learning. Despite the outstanding performance, the global features fail to preserve node-level embedding information adequately.

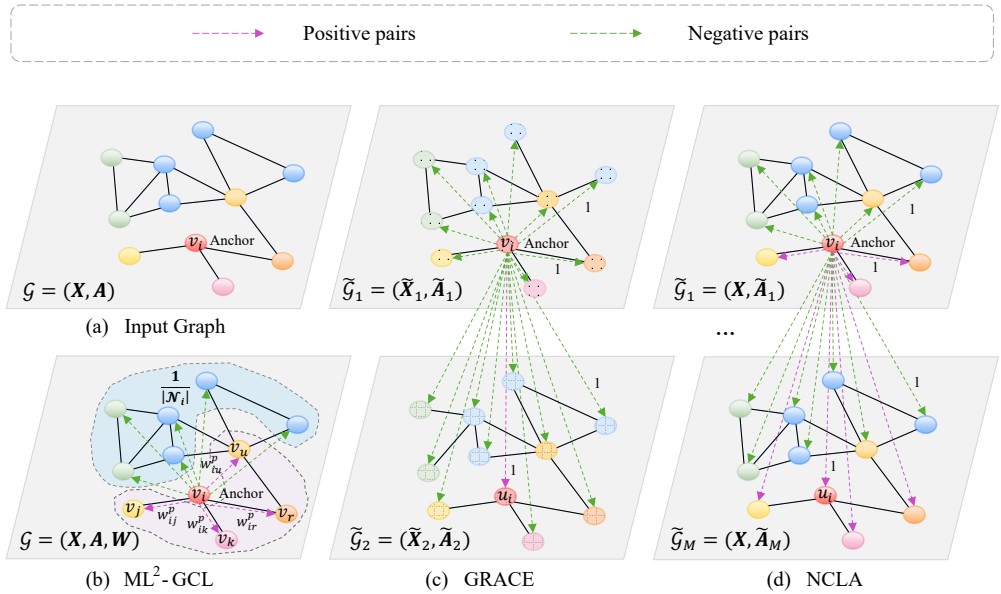

*Figure 2.* ML²-GCL versus GRACE and NCLA. ML²-GCL learns positive and negative pairs weights with the input graphs. GRACE removes edges and masks features to generate views of graphs. NCLA generates $M$ learnable augmented views with adaptive topology by multi-head GAT.

GRACE (Zhu et al., 2020) first utilizes the InfoNCE loss (Oord et al., 2018) to maximize the mutual information between positive pairs and minimize that of negative pairs under two augmented views. Subsequently, a series of methods have been proposed and applied to node classification (You et al., 2020; 2021; Xia et al., 2022b; Shen et al., 2023; Liang et al., 2025) and graph classification (Tan et al., 2021; Yin et al., 2022; Liu et al., 2023) tasks. While these methods have achieved significant success, generation of multiple views results in substantial computational overhead. Inspired by BYOL (Grill et al., 2020), BGRL (Thakoor et al., 2022) uses the random augmentation strategy to generate two views and regards the corresponding augmented nodes as positive pairs. Despite the improvement in graph representation learning and computational efficiency, random augmentation may disrupt the underlying semantics of graphs. AFGRL (Lee et al., 2022) generates two views with encoders and employs the $k$-means algorithm for positive pair sampling. The random initialization of the $k$-means algorithm may influence the performance.

## 2.2. Graph Augmentation

Graph augmentation strategies include node dropping (You et al., 2020), edge perturbation (Qiu et al., 2020; Zhang et al., 2023), attribute masking (Zhu et al., 2021; Zhang et al., 2022) and subgraph extraction (Hassani & Khasahmadi, 2020). GRACE (Zhu et al., 2020) uses random topology perturbation and node attribute masking to generate two augmented views. GCA (Zhu et al., 2021) proposes an adap-

tive augmentation strategy that integrates both structural and attribute information. CI-GCL (Tan et al., 2021) proposes a community-invariant GCL framework to preserve the community structure of graphs. Similarly, HomoGCL (Li et al., 2023) employs community detection algorithms to assess the importance of edges and node features. GCS (Wei et al., 2023) leverages both structural and semantic information to achieve adaptive graph augmentation. Although these methods have achieved significant success, existing augmentation strategies often exhibit poor adaptability across diverse graphs, which is due to the potential risks of augmentation strategies to disturb the underlying semantics. To avoid the above-mentioned issues, some works (Lee et al., 2022; Mo et al., 2022; Xia et al., 2022a; Liang et al., 2025) apply graph encoders to generate multiple views.

## 2.3. Node Sampling

Existing works on node sampling can be divided into two types, i.e., methods with graph structure relations and probabilistic models. Earlier works with graph structure relations (Zhu et al., 2020; You et al., 2020; 2021; Zhang et al., 2023) randomly perturb graphs to generate augmented views. They regard the corresponding augmented nodes as positive pairs and all the other nodes of augmented views as negative pairs. Therefore, each anchor point has only one positive pair. AFGRL (Lee et al., 2022) constructs multiple positive pairs from three aspects, including $k$-means clustering sets, $k$-NN sets and neighboring nodes. NCLA (Shen et al., 2023) directly utilizes the neighbors of anchor points as positive

pairs. GTCA (Liang et al., 2025) utilizes the node representation with GNN and Transformer as graph encoders and topological property, and thus shows excellent performance. While these methods have made some progress, multiple views may generate a large number of pairs, inevitably cause high computational complexity. HomoGCL (Li et al., 2023) regards $k$-means as a special case of the Gaussian Mixture Model to estimate the probability of positive pairs. NF-N2N (Dong et al., 2024) uses mutual information to measure the topological similarity between anchor nodes and their neighbors for positive pair sampling. SIGNA (Sun et al., 2024) randomly selects neighboring nodes as positive pairs. These methods may suffer from issues related to randomness and high complexity due to an excessive number of parameters.

# 3. Method

In this section, we first present the preliminaries and notations about GCL. Then, we propose ML²-GCL. Finally, we give the theoretical analysis and complexity analysis.

## 3.1. Preliminaries and Notations

Let $\mathcal{G} = (\mathcal{V}, \mathcal{E})$ be an input graph, where $\mathcal{V} = \{v_1, \cdots, v_N\}$, $\mathcal{E} \subseteq \mathcal{V} \times \mathcal{V}$ are the node set and the edge set, respectively. We denote the embedding matrix and the adjacency matrix as $\boldsymbol{X} \in \mathbb{R}^{N \times F}$ and $\boldsymbol{A} \in \{0,1\}^{N \times N}$, where $\boldsymbol{x}_i \in \mathbb{R}^F$ is the node feature of $v_i$, and $\boldsymbol{A}_{ij} = 1$ iff $(v_i, v_j) \in \mathcal{E}$. The goal of GCL is to learn a graph encoder $f_\theta(\boldsymbol{X}, \boldsymbol{A}) \in \mathbb{R}^{N \times D}$ and get node embeddings in low dimensional space without label information, i.e., $D \ll F$, which can be directly applied to downstream tasks.

Existing GCL methods typically generate positive/negative pairs with augmentation strategies. Take one of the most popular GCL methods GRACE (Zhu et al., 2020) as an example, it generates 2 augmented views $\tilde{\mathcal{G}}_1 = (\tilde{\boldsymbol{X}}_1, \tilde{\boldsymbol{A}}_1)$ and $\tilde{\mathcal{G}}_2 = (\tilde{\boldsymbol{X}}_2, \tilde{\boldsymbol{A}}_2)$. Then, it encodes the 2 views with the same GNN encoder to get node embeddings $\boldsymbol{Z} = f_\theta(\tilde{\boldsymbol{X}}_1, \tilde{\boldsymbol{A}}_1)$ and $\boldsymbol{Z}' = f_\theta(\tilde{\boldsymbol{X}}_2, \tilde{\boldsymbol{A}}_2)$. Finally, the loss function defined by the InfoNCE loss (Oord et al., 2018) is as follows

$$\mathcal{L} = \frac{1}{2N} \sum_{i=1}^{N} (\ell(\boldsymbol{z}_i, \boldsymbol{z}_i') + \ell(\boldsymbol{z}_i', \boldsymbol{z}_i)) \tag{1}$$

where

$$\ell(\boldsymbol{z}_i, \boldsymbol{z}_i') =$$
$$-\log \frac{\exp\left(s(\boldsymbol{z}_i, \boldsymbol{z}_i')/\tau\right)}{\underbrace{\exp\left(s(\boldsymbol{z}_i, \boldsymbol{z}_i')/\tau\right)}_{\text{positive pair}} + \underbrace{\sum_{j \neq i} \exp\left(s(\boldsymbol{z}_i, \boldsymbol{z}_j')/\tau\right)}_{\text{inter-view negative pairs}} + \underbrace{\sum_{j \neq i} \exp\left(s(\boldsymbol{z}_i, \boldsymbol{z}_j)/\tau\right)}_{\text{intra-view negative pairs}}} \tag{2}$$

where $s(\cdot, \cdot)$ is the similarity function and $\tau$ is a temperature parameter.

## 3.2. ML²-GCL

**Neighboring Sampling** As Figures 1-2 present, ML²-GCL is based on simple geometric intuitions. Suppose the anchor point and its $k$-hop neighboring nodes are sampled from the same underlying manifold. Assume there are sufficient nodes, i.e., the manifold is well-sampled, we expect each node and its neighbors to lie on or close to a locally linear patch of the manifold. We characterize the local geometry of these nodes by linear coefficients that reconstruct each anchor point from its positive pairs. Then, the optimization objective is as follows

$$\min_{w_{ij}^p} \sum_{i=1}^{N} ||\boldsymbol{x}_i - \sum_{v_j \in \mathcal{P}_i} w_{ij}^p \boldsymbol{x}_j||_2^2 + \lambda \sum_{i=1}^{N} \sum_{v_j \in \mathcal{P}_i} (w_{ij}^p)^2$$
$$s.t. \ w_{ij}^p \geq 0, i = 1, 2, ..., N, v_j \in \mathcal{P}_i \tag{3}$$
$$\sum_{v_j \in \mathcal{P}_i} w_{ij}^p = 1, i = 1, 2, ..., N$$

where $\mathcal{P}_i$ is the positive pair set of $v_i$. It consists of $k$-hop neighboring nodes of $v_i$. $w_{ij}^p$ is the reconstruction weight to be solved, which can also be regarded as the probability that $v_i$ and $v_j$ are positive pairs. $\lambda$ is a positive hyperparameter to balance the reconstruction error and the regularizer.

For simplicity, Equation 3 can be converted to the problem as following vector form

$$\min_{\boldsymbol{w}_i^p} \sum_{i=1}^{N} (||\boldsymbol{x}_i - \sum_{v_j \in \mathcal{P}_i} w_{ij}^p \boldsymbol{x}_j||_2^2 + \lambda ||\boldsymbol{w}_i^p||_2^2)$$
$$s.t. \ w_{ij}^p \geq 0, i = 1, 2, ..., N, v_j \in \mathcal{P}_i \tag{4}$$
$$\sum_{v_j \in \mathcal{P}_i} w_{ij}^p = 1, i = 1, 2, ..., N$$

Figure 1 shows the equivalent matrix form with $w_{ik}^p = 0, v_k \in \mathcal{N}_i$. The optimization problem above is a typical quadratic programming problem that is easy to solve. Refer to Sec 3.3 for detailed theoretical analysis.

Then, for a given anchor point $v_i$, we assign all of the negative pairs the same weight as

$$w_{ik}^n = \frac{1}{|\mathcal{N}_i|}, v_k \in \mathcal{N}_i \tag{5}$$

where $\mathcal{N}_i$ is the negative pair set of $v_i$. Similarly, we set $w_{ij}^n = 0, v_j \in \mathcal{P}_i$.

**Graph Encoders** We apply GCN (Kipf & Welling, 2017) as graph encoder. GCN uses a number of graph convolution

layers to aggregate information from neighbors and then updates each layer with the equation as follows

$$Z^{(l+1)} = \sigma \left( \hat{A} \, Z^{(l)} \, W^{(l)} \right) \qquad (6)$$

where $Z^{(l)}$ is the feature matrix at layer $l$, $Z^{(0)} = X$, $\hat{A}$ is the normalized adjacency matrix with self-loops, $W^{(l)}$ denotes the learnable weight matrix of layer $l$ and $\sigma$ is a non-linear activation function, i.e., ReLU.

**Overview**  As a lower bound of the Mutual Information, InfoNCE (Oord et al., 2018) is the most commonly used and popular loss function in GCL. In recent years, a large number of related works have sprung up (Zhu et al., 2020; 2021; Xia et al., 2022b; Zhang et al., 2023; Guo et al., 2023; Yu et al., 2024). In spite of some success, the exploited augmentation strategy may bring potential risks of the underlying semantics disturbance. Moreover, only one positive pair for an anchor point can inevitably exacerbate the sampling bias problem. To remedy these deficiencies, we construct multiple positive pairs for a given anchor point with its $k$-hop neighboring nodes and original node embeddings $X$ as Figures 1-2 show. Then, we use GCN as graph encoder to get the projected node embeddings $Z$. In the end, we define the training objective for each anchor point $v_i$ as follows

$$\ell(z_i) =$$
$$- \log \frac{\sum\limits_{v_j \in \mathcal{P}_i} w_{ij}^p \exp\left(s(z_i, z_j)/\tau\right)}{\sum\limits_{v_j \in \mathcal{P}_i} w_{ij}^p \exp\left(s(z_i, z_j)/\tau\right) + \sum\limits_{v_k \in \mathcal{N}_i} w_{ik}^n \exp\left(s(z_i, z_k)/\tau\right)}$$
$$(7)$$

where $\tau \in [0, 1]$ is a tunable temperature parameter to adjust the degree of attention to the hard negative samples.

Finally, we can define the following overall loss

$$\mathcal{L} = \frac{1}{N} \sum_{i=1}^{N} \ell(z_i) \qquad (8)$$

### 3.3. Theoretical Analysis

We propose the manifold learning inspired positive sampling strategy for GCL from the node neighbor sets. To our best knowledge, this is the first exploration to marry manifold learning with graph contrastive learning. It is superior to the similarity/distance-based approaches and probability models that previous GCL models adopt. The details of the proofs are provided in the Appendix A.

In the following, we will give the closed form solution of $w^p$ without constraints and with constraints, respectively.

**Proposition 1.** Equation 4 without constraints has an optimal closed solution $w_i^p = (X_i X_i^T + \lambda I)^{-1} X_i x_i$, where $w_i^p \in \mathbb{R}^{|\mathcal{P}_i|}$ is the reconstruction weight vector of $v_i$,

$X_i \in \mathbb{R}^{|\mathcal{P}_i| \times F}$ is the positive pairs embedding matrix of $v_i$ and $x_i \in \mathbb{R}^F$ is the embedding vector of $v_i$.

**Proposition 2.** The optimal closed solution of Equation 4 can be derived with non-negativity and normalization constraints on solution of Proposition 1. Specifically, we may input the weights into the ReLU activation function, i.e., $w_i^p = \text{ReLU}(w_i^p)$ and normalize the weights, i.e., $w_i^p = w_i^p / \sum\limits_{j=1}^{N} w_{ij}^p$.

In fact, the essence of GCL is instance discrimination as a pretext task for self-supervised learning (Wang et al., 2022; Liu et al., 2021). However, learning semantics relies heavily on intra-class data distribution, as shown in (Wang et al., 2022). It is limited by the one-to-one positive sampling strategy, and ML²-GCL may improve it to form a new task that we refer to as **personalized graph embedding** task, which has less reliance on the data distribution.

**Theorem 1.** Suppose for a given anchor point $v_i$ in a graph, other nodes can be divided into its positive pairs or negative pairs. The training objective is equivalent to graph embedding as given:

$$\ell \iff \frac{\sum_{v_k \in \mathcal{N}_i} w_{ik}^n s(z_i, z_k)}{\sum_{v_j \in \mathcal{P}_i} w_{ij}^p s(z_i, z_j)} \qquad (9)$$

We present the whole process of our ML²-GCL method in Algorithm 1.

---

**Algorithm 1** ML²-GCL

---
**Input**: Graph $\mathcal{G} = (\mathcal{V}, \mathcal{E})$
**Output**: Node embeddings $Z$

1: **for** i in 1 to $N$ **do**
2:    Construct positive pair set $\mathcal{P}_i$ of $v_i$ with its $k$-hop neighboring nodes;
3:    Construct negative pair set $\mathcal{N}_i$ of $v_i$ with other nodes;
4: **end for**
5: Compute positive pairs weight matrix $W^p$ as in Propositions 1-2;
6: Compute negative pairs weight matrix $W^n$ as in Equation 5;
7: Initialize GNN encoder parameters $\{W, b\}$;
8: **while** not converge **do**
9:    Obtain GNN embeddings $Z = f_\theta(\mathcal{G})$;
10:    Do forward pass, compute $\mathcal{L}$ as in Equations 7-8;
11:    Do backward propagation with $\mathcal{L}$;
12: **end while**
13: **return** $Z$ for downstream tasks;

---

### 3.4. Complexity Analysis

Consider a graph with $N$ nodes, $M$ edges and simple encoders which compute embeddings in time and space

*Table 1.* Node classification accuracy (%) on 6 datasets. Best results are colored: **first**, **second**, **third**. Models with stars adopt graph augmentation strategy. Models in italic type use at least 2 views.

| Model | Cora | Citeseer | Pubmed | Amazon-Photo | Amazon-Computers | Wiki-CS |
|---|---|---|---|---|---|---|
| GCN (Kipf & Welling, 2017) | $79.6 \pm 1.8$ | $66.0 \pm 1.2$ | $79.0 \pm 2.5$ | $86.3 \pm 1.6$ | $76.4 \pm 1.8$ | $67.3 \pm 1.5$ |
| GAT (Velickovic et al., 2018) | $81.2 \pm 1.6$ | $68.9 \pm 1.8$ | $78.5 \pm 1.8$ | $86.5 \pm 2.1$ | $77.9 \pm 1.8$ | $68.6 \pm 1.9$ |
| *CGPN* (Wan et al., 2021b) | $74.0 \pm 1.7$ | $63.7 \pm 1.6$ | $73.3 \pm 2.5$ | $84.1 \pm 1.5$ | $74.7 \pm 1.3$ | $66.1 \pm 2.1$ |
| *CG3* (Wan et al., 2021a) | $80.6 \pm 1.6$ | $70.9 \pm 1.5$ | $78.9 \pm 2.6$ | $89.4 \pm 1.9$ | $77.8 \pm 1.7$ | $68.0 \pm 1.5$ |
| *DGI*[*] (Velickovic et al., 2019) | $82.1 \pm 1.3$ | $71.6 \pm 1.2$ | $78.3 \pm 2.4$ | $83.5 \pm 1.2$ | $78.8 \pm 1.1$ | $69.1 \pm 1.4$ |
| *GMI* (Peng et al., 2020) | $79.4 \pm 1.2$ | $66.9 \pm 2.2$ | $76.8 \pm 2.3$ | $86.7 \pm 1.5$ | $76.1 \pm 1.2$ | $67.8 \pm 1.8$ |
| *MVGRL*[*] (Hassani & Khasahmadi, 2020) | $82.4 \pm 1.5$ | $71.1 \pm 1.4$ | $79.5 \pm 2.2$ | $89.7 \pm 1.2$ | $78.7 \pm 1.7$ | $69.2 \pm 1.2$ |
| *GRACE*[*] (Zhu et al., 2020) | $79.6 \pm 1.4$ | $67.0 \pm 1.7$ | $74.6 \pm 3.5$ | $87.9 \pm 1.4$ | $76.8 \pm 1.7$ | $67.8 \pm 1.4$ |
| *GCA*[*] (Zhu et al., 2021) | $79.0 \pm 1.4$ | $65.6 \pm 2.4$ | $81.5 \pm 2.5$ | $87.0 \pm 1.9$ | $76.9 \pm 1.4$ | $67.6 \pm 1.3$ |
| *SUGRL* (Mo et al., 2022) | $81.3 \pm 1.2$ | $71.0 \pm 1.8$ | $80.5 \pm 1.6$ | $90.5 \pm 1.9$ | $78.2 \pm 1.2$ | $68.7 \pm 1.1$ |
| *AFGRL* (Lee et al., 2022) | $78.6 \pm 1.3$ | $70.8 \pm 2.1$ | $76.4 \pm 2.5$ | $89.2 \pm 1.1$ | $77.7 \pm 1.1$ | $68.0 \pm 1.7$ |
| *NCLA* (Shen et al., 2023) | $82.2 \pm 1.6$ | $71.7 \pm 0.9$ | $82.0 \pm 1.4$ | $90.2 \pm 1.3$ | $79.8 \pm 1.5$ | $70.3 \pm 1.7$ |
| *GTCA* (Liang et al., 2025) | $82.5 \pm 1.3$ | $69.7 \pm 1.7$ | $79.8 \pm 1.3$ | $90.5 \pm 1.2$ | $79.2 \pm 1.4$ | $69.7 \pm 1.5$ |
| ML²-GCL | $83.0 \pm 1.3$ | $71.7 \pm 1.0$ | $82.1 \pm 1.8$ | $90.7 \pm 1.6$ | $83.0 \pm 2.1$ | $73.3 \pm 1.7$ |

$O(M + N)$. This property is satisfied by most popular GNN architectures, such as GCN (Kipf & Welling, 2017) and GAT (Velickovic et al., 2018). In each update step, ML²-GCL performs 1 encoder computation and backpropagates the learning signal once plus a prediction step, while GRACE performs 2 encoder computations (once for each augmentation) and backpropagates the learning signal twice (once for each augmentation), plus a prediction step. We assume the backward pass to be approximately as costly as the forward pass. We ignore the cost of augmentation computation in this analysis. Therefore, the total time and space complexity per update step for ML²-GCL is $2C_{\text{encoder}}(M + N) + 2C_{\text{prediction}}N + CN^2$, compared to $4C_{\text{encoder}}(M + N) + 4C_{\text{prediction}}N + 4CN^2$ for GRACE, where $C$ are constants dependent on architecture of different components. Appendix D gives further details.

## 4. Experiments

In this section, we conduct a series of experiments to demonstrate the superiority of ML²-GCL. First, we briefly describe the datasets. Then, we evaluate the empirical performance across various graph datasets on node classification and link prediction tasks. In the end, we present the ablation study, hyperparameters analysis and embeddings visualization results. We implement all experiments on the platform with PyTorch 2.0.1 and PyTorch Geometric 2.6.1 on NVIDIA 3090 GPUs with 24GB memory.

### 4.1. Datasets

We conduct experiments on 6 widely-used datasets including Cora, Citeseer, Pubmed, Amazon-Photo, Amazon-Computers and Wiki-CS. For node classification task, we split Cora, Citeseer and Pubmed for the training, validation

and testing following (Yang et al., 2016), and all the other datasets following (Liu et al., 2020). For link prediction task, we follow the experimental setup of GCA (Zhu et al., 2021). For each dataset, we conduct 20 random splits of training/validation/test, and report the average performance of all algorithms on the same random splits. The statistics are summarized in Appendix B.

### 4.2. Node Classification

Node classification is one of the common tasks in graph neural networks. The goal is to predict the class of each node according to its characteristics and context information in the graph structure. In this task, the graph structure and node embeddings are jointly used as inputs, and the relations between nodes and local or global topological information are captured through graph neural networks.

To evaluate node classification, we compare ML²-GCL with 13 state-of-the-art methods including 2 semi-supervised GNNs, i.e., GCN (Kipf & Welling, 2017), GAT (Velickovic et al., 2018), 2 semi-supervised GCL methods, i.e., CGPN (Wan et al., 2021b), CG3 (Wan et al., 2021a), and 9 self-supervised GCL methods, i.e., DGI (Velickovic et al., 2019), GMI (Peng et al., 2020), MVGRL (Hassani & Khasahmadi, 2020), GRACE (Zhu et al., 2020), GCA (Zhu et al., 2021), SUGRL (Mo et al., 2022), AFGRL (Lee et al., 2022), NCLA (Shen et al., 2023) and GTCA (Liang et al., 2025). For chosen hyper-parameters see Appendix C.

Table 1 lists the node classification accuracy of ML²-GCL and baselines with a logistic regression model. The results indicate that ML²-GCL achieves state-of-the-art results with respect to previous methods, especially when there are relatively more classes. For example, ML²-GCL achieves 83.0% and 73.3% accuracy on Amazon-Computers and Wiki-CS,

*Table 2.* Link prediction results (%) on 6 datasets. Best results are colored: first, second, third. Models with stars adopt graph augmentation strategy. Models in italic type use at least 2 views.

| Model | Cora | | Citeseer | | Pubmed | | Amazon-Photo | | Amazon-Computers | | Wiki-CS | |
|---|---|---|---|---|---|---|---|---|---|---|---|---|
| | AUC | AP | AUC | AP | AUC | AP | AUC | AP | AUC | AP | AUC | AP |
| Spectral (Ng et al., 2001) | 84.6 | 88.5 | 80.5 | 85 | 84.2 | 87.8 | 83.6 | 84.3 | 86.7 | 87.3 | 81.2 | 80.3 |
| DeepWalk (Perozzi et al., 2014) | 83.1 | 85 | 80.5 | 83.6 | 84.4 | 84.1 | 83.2 | 85.1 | 87.2 | 87.5 | 82.1 | 81.5 |
| GAE (Schulman et al., 2016) | 91 | 92 | 89.5 | 89.9 | 96.4 | 96.5 | 89.9 | 89.6 | 93 | 93.2 | 83.2 | 82.1 |
| VGAE (Kipf & Welling, 2016) | 91.4 | 92.6 | 90.8 | 92 | 94.4 | 94.7 | 89.3 | 88.8 | 92.6 | 92.8 | 82.5 | 83.4 |
| ARGE (Pan et al., 2018) | 92.4 | 93.2 | 91.9 | 93 | 96.8 | 97.1 | 91.5 | 92.2 | 92.5 | 92.7 | 83.5 | 84.2 |
| ARVGA (Pan et al., 2018) | 92.4 | 92.6 | 92.4 | 93 | 96.5 | 96.8 | 92.1 | 91.9 | 93.1 | 92.8 | 82.6 | 83.9 |
| *GRACE** (Zhu et al., 2020) | 90.9 | 91 | 92.1 | 92.2 | 97 | 97.1 | 90.8 | 89.3 | 91.6 | 91.2 | 89.5 | 88.5 |
| *GCA** (Zhu et al., 2021) | 91.4 | 91.5 | 92 | 92.6 | 96.3 | 96.5 | 92.3 | 91.3 | 92.5 | 91.5 | 86.4 | 86.3 |
| *GDCL** (Zhao et al., 2021) | 91.7 | 90.9 | 91.9 | 92 | 96.5 | 96.3 | 93.1 | 93.6 | 93.1 | 93.2 | 85.2 | 84.6 |
| *ProGCL** (Xia et al., 2022b) | 92.9 | 93.5 | 93.1 | 93.3 | 96.1 | 96.7 | 92.6 | 93.5 | 94.5 | 94.2 | 83.6 | 84.1 |
| *AUGCL** (Niu et al., 2024) | 93.3 | 93.2 | 92.5 | 92.8 | 96.3 | 96.5 | 94.2 | 93.9 | 93.7 | 93.9 | 88.9 | 88.5 |
| ML²-GCL | 96.9 | 95.9 | 96 | 95.1 | 98.7 | 98.5 | 95.1 | 94.2 | 96.9 | 96.6 | 89.8 | 88.9 |

*Table 3.* Ablation study on node classification task for positive/negative pairs weights. The metric is the Accuracy(%).

| $W^p$ | $W^n$ | Cora | Citeseer | Pubmed | Amazon-Photo | Amazon-Computers | Wiki-CS |
|---|---|---|---|---|---|---|---|
| – | – | $81.2 \pm 1.3$ | $71.0 \pm 0.9$ | $79.5 \pm 2.0$ | $89.5 \pm 1.4$ | $81.1 \pm 2.4$ | $70.0 \pm 1.9$ |
| – | ✓ | $81.9 \pm 1.3$ | $71.2 \pm 0.9$ | $80.1 \pm 1.7$ | $89.6 \pm 2.0$ | $82.0 \pm 2.7$ | $72.2 \pm 1.9$ |
| ✓ | – | $82.5 \pm 1.2$ | $71.7 \pm 1.1$ | $79.8 \pm 1.8$ | $90.4 \pm 1.2$ | $82.9 \pm 2.1$ | $72.5 \pm 2.2$ |
| ✓ | ✓ | $\mathbf{83.0 \pm 1.3}$ | $\mathbf{71.7 \pm 1.0}$ | $\mathbf{82.1 \pm 1.8}$ | $\mathbf{90.7 \pm 1.6}$ | $\mathbf{83.0 \pm 2.1}$ | $\mathbf{73.3 \pm 1.7}$ |

a 3.2% and 3.0% relative improvement over previous state-of-the-art respectively.

## 4.3. Link Prediction

Link prediction aims to predict whether there is an edge between two nodes in graphs, or to predict the weights of edges. The goal of this task is to capture potential relations between nodes through learning node representations. It can be widely applied in fields such as social network analysis, protein-protein interaction prediction in biological networks, and recommendation systems.

For link prediction, we consider our ML²-GCL with multiple baselines including Spectral (Ng et al., 2001), DeepWalk (Perozzi et al., 2014), GAE (Schulman et al., 2016), VGAE (Kipf & Welling, 2016), ARGE (Pan et al., 2018), ARVGA (Pan et al., 2018), GRACE (Zhu et al., 2020), GCA (Zhu et al., 2021), GDCL (Zhao et al., 2021), ProGCL (Xia et al., 2022b) and AUGCL (Niu et al., 2024). We utilize the Area Under Curve (AUC) and Average Precision (AP) to evaluate the performance. For chosen hyper-parameters see Appendix C.

The results shown in Table 2 suggest that ML²-GCL significantly outperforms all the other baselines on 6 datasets under different evaluation criteria, highlighting its potential on link prediction task. For example, it achieves 96.9% AUC

on Cora, i.e., a 3.6% relative improvement over previous state-of-the-art.

It is noteworthy that ML²-GCL achieves the state-of-the-art results on both node classification and link prediction benchmarks with a unified approach. Unlike previous unsupervised models, we do not devise a specialized encoder for each task.

## 4.4. Ablation Study

In this section, we remove positive/negative pairs weights of ML²-GCL to study the impact of each component. The results are shown in Table 3. We notice that the removal of either positive pairs weights $W^p$ or negative pairs weights $W^n$ leads to poorer performance, which demonstrates the effectiveness of our proposed ML²-GCL. In addition, it is obvious that compared with the negative pairs weights $W^n$, the positive pairs weights $W^p$ is more crucial for ML²-GCL, which is mainly due to its role on recovery of global nonlinear structure.

## 4.5. Hyperparameters Analysis

Figure 3 illustrates the node classification accuracy of ML²-GCL with varying values of the temperature parameter $\tau$ and the hidden dim on 6 datasets. It is noteworthy that the optimal temperature parameter is around 0.5 on most of the

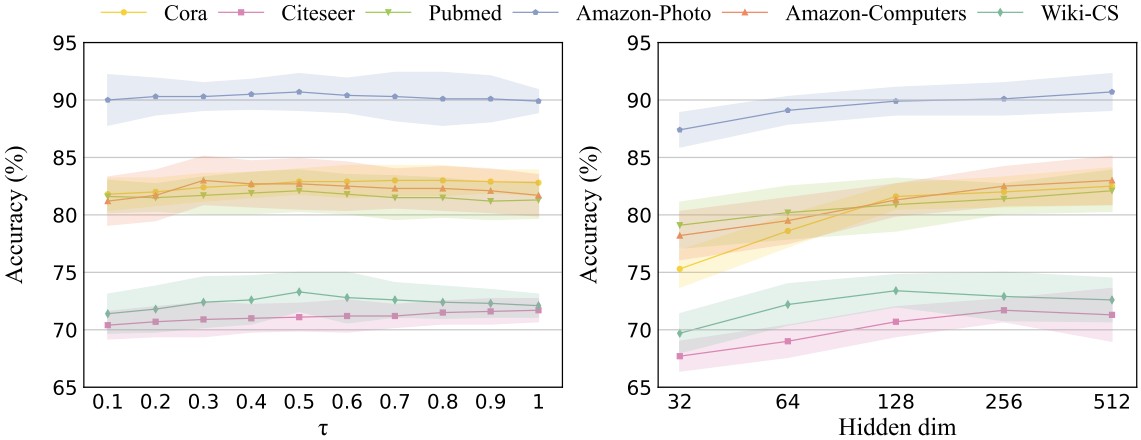

*Figure 3.* Sensitivity analysis of the temperature parameter $\tau$ and the hidden dim on node classification task.

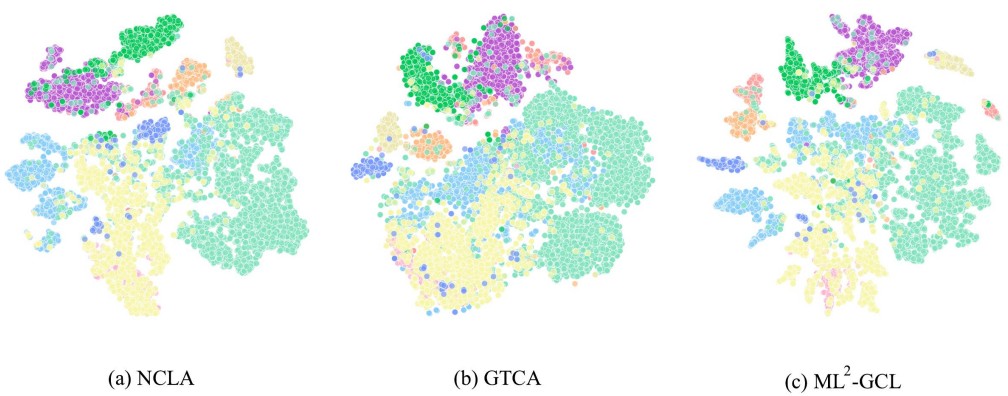

(a) NCLA                    (b) GTCA                    (c) ML$^2$-GCL

*Figure 4.* Visualization of NCLA, GTCA and ML$^2$-GCL embeddings on Amazon-Computers dataset with t-SNE.

datasets. In general, with the increase of the hidden dim, the classification performance shows an upward trend on most of the datasets such as Pubmed, Amazon-Photo and Amazon-Computers. We notice that ML$^2$-GCL achieves the best performance on Wiki-CS when the hidden dim is 128. This is mainly due to the fact that the original feature dimension of Wiki-CS is only 300. Increasing the hidden dim will introduce irrelevant or redundant information.

### 4.6. Embeddings Visualization

To provide a more intuitive presentation of the node embeddings, we utilize t-SNE (Van der Maaten & Hinton, 2008) to visualize the node embeddings of NCLA (Shen et al., 2023), GTCA (Liang et al., 2025) and ML$^2$-GCL on Amazon-Computers dataset. As Figure 4 shows, different colors represent different classes. It is obvious that ML$^2$-GCL can distinguish much more different classes of nodes effectively compared with NCLA and GTCA.

## 5. Conclusion

In this paper, we marry manifold learning with graph contrastive learning and develop ML$^2$-GCL. We achieve global nonlinear structure recovery from locally linear fits, and thus inherit the merits of both manifold learning and GCN. It allows us to develop a new paradigm of contrastive learning to eliminate extra views generation and pairwise distances calculation, thus avoiding the semantic disturbance and high computational complexity completely. In addition, we conduct positive sampling with the optimal closed-form solution of a typical quadratic programming problem. Theoretical analysis proves its existence. Extensive experiments verify both effectiveness and lightweight of ML$^2$-GCL. In future, we will investigate deep integration mechanisms between manifold learning and dynamic graph structures to address nonlinear evolution patterns in temporal data or dynamic networks.

## Acknowledgements

This work is supported by National Natural Science Foundation of China (No.62376142, U21A20473).

## Impact Statement

This paper presents work whose goal is to advance the field of Machine Learning. There are many potential societal consequences of our work, none which we feel must be specifically highlighted here.

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

# A. Theoretical Proofs

## A.1. Proof of Proposition 1

**Proposition 1.** Equation 4 without constraints has an optimal closed solution $\boldsymbol{w}_i^p = (\boldsymbol{X}_i \boldsymbol{X}_i^T + \lambda \boldsymbol{I})^{-1} \boldsymbol{X}_i \boldsymbol{x}_i$, where $\boldsymbol{w}_i^p \in \mathbb{R}^{|\mathcal{P}_i|}$ is the reconstruction weight vector of $v_i$, $\boldsymbol{X}_i \in \mathbb{R}^{|\mathcal{P}_i| \times F}$ is the positive pairs embedding matrix of $v_i$ and $\boldsymbol{x}_i \in \mathbb{R}^F$ is the embedding vector of $v_i$.

*Proof.* First, we review Equation 4 of our proposed ML²-GCL method and rewrite the objective for each anchor point $v_i, i = 1, 2, ..., N$ without constraints as:

$$l_i(\boldsymbol{w}_i^p) = ||\boldsymbol{x}_i - \sum_{v_j \in \mathcal{P}_i} w_{ij}^p \boldsymbol{x}_j||_2^2 + \lambda ||\boldsymbol{w}_i^p||_2^2 \tag{10}$$

Then, we take the derivative of $l_i$ with respect to $\boldsymbol{w}_i^p$

$$\frac{\mathrm{d}l_i}{\mathrm{d}\boldsymbol{w}_i^p} = -2\boldsymbol{X}_i^T \boldsymbol{x}_i + 2\boldsymbol{X}_i^T \boldsymbol{X}_i \boldsymbol{w}_i^p + 2\lambda \boldsymbol{w}_i^p \tag{11}$$

Let $\frac{\mathrm{d}l_i}{\mathrm{d}\boldsymbol{w}_i^p} = 0$, we obtain

$$(\boldsymbol{X}_i^T \boldsymbol{X}_i + \lambda \boldsymbol{I})\boldsymbol{w}_i^p = \boldsymbol{X}_i^T \boldsymbol{x}_i \tag{12}$$

Considering $\boldsymbol{X}_i^T \boldsymbol{X}_i \succeq \boldsymbol{0}$ and $\lambda > 0$, we may conclude the second derivative of $l_i$ with respect to $\boldsymbol{w}_i^p$

$$\frac{\mathrm{d}^2 l_i}{\mathrm{d}(\boldsymbol{w}_i^p)^2} = 2\boldsymbol{X}_i^T \boldsymbol{X}_i + 2\lambda \boldsymbol{I} \succ \boldsymbol{0} \tag{13}$$

Hence, $\boldsymbol{w}_i^p = (\boldsymbol{X}_i^T \boldsymbol{X}_i + \lambda \boldsymbol{I})^{-1} \boldsymbol{X}_i^T \boldsymbol{x}_i$ is the extreme point of Equation 10 and also the optimal solution of Equation 4. $\square$

## A.2. Proof of Proposition 2

**Proposition 2.** The optimal closed solution of Equation 4 can be derived with non-negativity and normalization constraints on solution of Proposition 1. Specifically, we may input the weights into the ReLU activation function, i.e., $\boldsymbol{w}_i^p = \mathrm{ReLU}(\boldsymbol{w}_i^p)$ and normalize the weights, i.e., $\boldsymbol{w}_i^p = \boldsymbol{w}_i^p / \sum_{j=1}^N w_{ij}^p$.

*Proof.* As $\mathrm{ReLU}(\boldsymbol{w}_i^p) = \max(\boldsymbol{0}, \boldsymbol{w}_i^p)$, we may infer that $w_{ij}^p = \max(0, w_{ij}^p) \geq 0, i = 1, 2, ..., N, v_j \in \mathcal{P}_i$. In addition, we can derive conclusion as follows

$$\sum_{v_j \in \mathcal{P}_i} w_{ij}^p = \sum_{v_j \in \mathcal{P}_i} \frac{w_{ij}^p}{\sum_{j=1}^N w_{ij}^p} = \frac{\sum_{v_j \in \mathcal{P}_i} w_{ij}^p}{\sum_{v_j \in \mathcal{P}_i} w_{ij}^p + \sum_{v_k \in \mathcal{N}_i} w_{ik}^p} = 1, i = 1, 2, ..., N \tag{14}$$

where $w_{ik}^p = 0, v_k \in \mathcal{N}_i$. $\square$

## A.3. Proof of Theorem 1

**Theorem 1.** Suppose for a given anchor point $v_i$ in a graph, other nodes can be divided into its positive pairs or negative pairs. The training objective is equivalent to graph embedding as given:

$$\ell \iff \frac{\sum_{v_k \in \mathcal{N}_i} w_{ik}^n s(\boldsymbol{z}_i, \boldsymbol{z}_k)}{\sum_{v_j \in \mathcal{P}_i} w_{ij}^p s(\boldsymbol{z}_i, \boldsymbol{z}_j)} \tag{15}$$

*Proof.* We rewrite the loss function of anchor point $v_i$ with Taylor expansion of the first order as:

$$
\begin{aligned}
\ell(\boldsymbol{z}_i) &= \log \frac{\sum\limits_{v_j \in \mathcal{P}_i} w_{ij}^p \exp\left(s(\boldsymbol{z}_i, \boldsymbol{z}_j)/\tau\right) + \sum\limits_{v_k \in \mathcal{N}_i} w_{ik}^n \exp\left(s(\boldsymbol{z}_i, \boldsymbol{z}_k)/\tau\right)}{\sum\limits_{v_j \in \mathcal{P}_i} w_{ij}^p \exp\left(s(\boldsymbol{z}_i, \boldsymbol{z}_j)/\tau\right)} \\
&= \log\left(1 + \frac{\sum\limits_{v_k \in \mathcal{N}_i} w_{ik}^n \exp\left(s(\boldsymbol{z}_i, \boldsymbol{z}_k)/\tau\right)}{\sum\limits_{v_j \in \mathcal{P}_i} w_{ij}^p \exp\left(s(\boldsymbol{z}_i, \boldsymbol{z}_j)/\tau\right)}\right) \\
&\approx \frac{\sum\limits_{v_k \in \mathcal{N}_i} w_{ik}^n \exp\left(s(\boldsymbol{z}_i, \boldsymbol{z}_k)/\tau\right)}{\sum\limits_{v_j \in \mathcal{P}_i} w_{ij}^p \exp\left(s(\boldsymbol{z}_i, \boldsymbol{z}_j)/\tau\right)}
\end{aligned}
\tag{16}
$$

We extend the loss function of $v_i$ to the form of the entire graph $\mathcal{G}$ and remove the temperature hyperparamter $\tau$:

$$
\begin{aligned}
\mathcal{L} &= \frac{1}{N} \sum_{i=1}^{N} \ell(\boldsymbol{z}_i) \\
&\propto \sum_{i=1}^{N} \frac{\sum\limits_{v_k \in \mathcal{N}_i} w_{ik}^n \exp\left(s(\boldsymbol{z}_i, \boldsymbol{z}_k)\right)}{\sum\limits_{v_j \in \mathcal{P}_i} w_{ij}^p \exp\left(s(\boldsymbol{z}_i, \boldsymbol{z}_j)\right)}
\end{aligned}
\tag{17}
$$

Different from typical graph embedding, each positive pair or negative pair is associated with its own weight. The overall objective is calculated with the exponential ratios sum of the negative and positive similarities with anchor points. □

# B. Dataset Descriptions

We evaluate the performance of ML²-GCL on node-level and edge-level tasks, i.e., node classification and link prediction. We conduct experiments on 6 widely used datasets including Cora, Citeseer, Pubmed, Amazon-Photo, Amazon-Computers and Wiki-CS. The detailed statistics are summarized in Table 4.

- **Cora, Citeseer** and **PubMed** (Sen et al., 2008) are well-known citation network datasets, where nodes represent scientific papers and edges represent citation relations. Node features are bag-of-words vectors of papers, and labels represent domains of papers.

- **Amazon-Photo** and **Amazon-Computers** (Shchur et al., 2018) are two networks of co-purchase relationships constructed from Amazon. Nodes are product and edges exist when two products are frequently co-purchased. Each node has a bag-of-words feature encoding product reviews and is labeled with its product category.

- **Wiki-CS** (Mernyei & Cangea, 2020) is a reference network from Wikipedia references. Nodes correspond to articles about computer science and edges are hyperlinks between the articles. Articles are labeled with 10 related subfields, and their features are calculated as the average of pre-trained Glove (Pennington et al., 2014) word embeddings.

*Table 4.* Statistics of datasets used in experiments.

| Dataset | # Nodes | # Edges | # Features | # Labels |
|---|---|---|---|---|
| Cora | 2,708 | 5,429 | 1,433 | 7 |
| Citeseer | 3,327 | 4,732 | 3,703 | 6 |
| Pubmed | 19,717 | 44,338 | 500 | 3 |
| Amazon-Photo | 7,650 | 119,081 | 745 | 8 |
| Amazon-Computers | 13,752 | 245,861 | 767 | 10 |
| Wiki-CS | 11,701 | 216,123 | 300 | 10 |

## C. Hyperparameter Choices

In hyperparameter search, we attempt to adjust the value of $k$ and $\lambda$ in ML²-GCL, as well as other deep learning hyperparameters including temperature parameter $\tau$, hidden dim, learning rate, dropout and weight decay. We apply the grid search strategy to choose the optimal hyperparameters. Specifically, we search $k$ in [1, 5], $\lambda$ in {0, 0.01, 0.1, 1, 10, 100}, hidden dim from {128, 256, 500, 512}, learning rate from {0.0005, 0.001, 0.005, 0.01} and weight decay in {5e-5, 1e-4, 5e-4, 7e-4}. Tables 5-6 give the hyperparameters specifications for ML²-GCL on node classification and link prediction tasks.

*Table 5.* Hyperparameters specifications for ML²-GCL on node classification task.

|  | Cora | Citeseer | Pubmed | Amazon-Photo | Amazon-Computers | Wiki-CS |
|---|---|---|---|---|---|---|
| $k$ | 1 | 1 | 1 | 1 | 1 | 1 |
| $\lambda$ | 0.01 | 1 | 100 | 10 | 0.01 | 10 |
| Temperature $\tau$ | 0.7 | 1 | 0.5 | 0.5 | 0.3 | 0.5 |
| # Hidden dim | 500 | 256 | 512 | 512 | 512 | 128 |
| Learning rate | 0.0005 | 0.005 | 0.001 | 0.001 | 0.0005 | 0.01 |
| # Epochs | 40 | 20 | 80 | 120 | 240 | 280 |
| Dropout | 0.5 | 0.4 | 0.5 | 0.3 | 0.5 | 0.4 |
| Weight decay | 5e-5 | 7e-4 | 5e-4 | 5e-4 | 5e-4 | 1e-4 |

*Table 6.* Hyperparameters specifications for ML²-GCL on link prediction task.

|  | Cora | Citeseer | Pubmed | Amazon-Photo | Amazon-Computers | Wiki-CS |
|---|---|---|---|---|---|---|
| $k$ | 3 | 3 | 2 | 2 | 4 | 1 |
| $\lambda$ | 10 | 0.001 | 1 | 100 | 100 | 1 |
| Temperature $\tau$ | 0.4 | 0.5 | 0.3 | 0.4 | 0.3 | 0.3 |
| # Hidden dim | 500 | 256 | 500 | 500 | 256 | 500 |
| Learning rate | 0.0005 | 0.01 | 0.0001 | 0.0001 | 0.01 | 0.001 |
| # Epochs | 200 | 100 | 200 | 100 | 200 | 40 |
| Dropout | 0.4 | 0.4 | 0.1 | 0.1 | 0.1 | 0.1 |
| Weight decay | 5e-5 | 5e-5 | 5e-5 | 1e-4 | 5e-5 | 5e-5 |

## D. Memory Usage

Table 7 shows the GPU memory of ML²-GCL and baselines including GRACE, GCA, AFGRL, NCLA and GTCA on node classification task. The results indicate that ML²-GCL has less memory requirement compared with existing methods due to its closed-form solution of positive pairs weights and removal of pairwise distances calculation. Cross-view contrastive methods such as GRACE require additional storage for the embedding matrices of multiple augmented views, as well as the indices of positive and negative pairs, thereby increasing memory requirement.

*Table 7.* The GPU memory (GiBs) usage across 6 datasets on node classification task. Best results are colored: first, second, third.

| Model | Cora | Citeseer | Pubmed | Amazon-Photo | Amazon-Computers | Wiki-CS |
|---|---|---|---|---|---|---|
| GRACE | 0.47 | 0.55 | 13.17 | 2.11 | 6.23 | 4.46 |
| GCA | 0.58 | 0.69 | 14.32 | 2.34 | 6.61 | 5.01 |
| AFGRL | 0.37 | 0.48 | 12.59 | 2.12 | 5.94 | 4.41 |
| NCLA | 0.67 | 1.02 | 14.56 | 2.25 | 7.16 | 7.71 |
| GTCA | 1.64 | 2.64 | 22.03 | 4.09 | 13.14 | 10.89 |
| ML²-GCL | 0.33 | 0.45 | 12.15 | 2.01 | 5.74 | 4.22 |

We show part of the information from Table 7 as a scatterplot to visualize the different scaling properties of ML²-GCL and

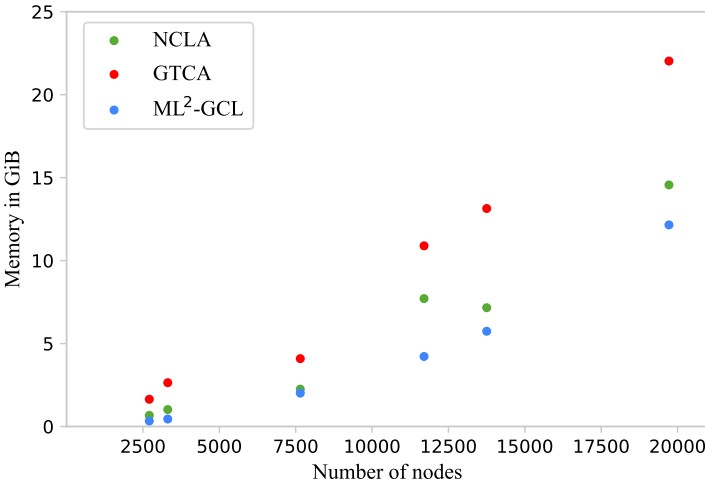

*Figure 5.* Memory usage of NCLA, GTCA and ML$^2$-GCL across 6 standard datasets on node classification task.

NCLA, GTCA intuitively in Figure 5 . Among the top 3 ranked methods on node classification task, ML$^2$-GCL consumes the least GPU memory usage. As the number of nodes increases, its memory usage increases slower compared with NCLA and GTCA.

