# OpenReview forum: "ML$^2$-GCL: Manifold Learning Inspired Lightweight Graph Contrastive Learning"
_ICML.cc/2025/Conference — ICML 2025 poster_

### Official Review · Reviewer_bpNM · 2025-03-10

**Overall Recommendation:** 4

**Summary:**

This paper proposes a lightweight graph contrastive learning (GCL) framework that integrates manifold learning theory, i.e., Manifold Learning Inspired Lightweight Graph Contrastive Learning (ML^2-GCL), aiming to optimize embedding representations through geometric structural constraints while reducing the computational complexity of traditional GCL methods. The method demonstrates innovative design, supported by rigorous theoretical analysis and extensive experimental validation.

**Claims And Evidence:**

The claims are well-supported by both theoretical analysis and extensive experiments.

**Essential References Not Discussed:**

To my best knowledge, no essential related works are missing.

**Experimental Designs Or Analyses:**

I checked the soundness of the experimental designs and analyses.

**Methods And Evaluation Criteria:**

The proposed ML^2-GCL method is both effective and lightweight for the problem of graph contrastive learning.

**Other Comments Or Suggestions:**

Have the authors tried other types of models as graph encoders, such as Graph Transformers?

**Other Strengths And Weaknesses:**

Strengths:
1. This approach aligns with recent trends in geometric property modeling of graph structures, emphasizing the intrinsic low-dimensional manifold characteristics of high-dimensional data. To my best knowledge, ML^2-GCL is the first to marry manifold learning with graph contrastive learning.
2. This paper has a clear research motivation and logical structure. It is well-written and easy to follow.
3. Solid theoretical analysis and extensive experimental results demonstrate the effectiveness and light weight of ML^2 -GCL.

Weaknesses:
1. The first step of ML^2-GCL is the neighboring sampling, which includes anchor node reconstruction combination weights, and then the authors apply the graph encoder. Such procedure is different from general GCL. The authors should give more detailed and reasonable explanation.
2. There are many numbers or symbols in the charts that are not displayed correctly. The author should check and modify them carefully.
3. There are some suggestions for grammar revision and improvement:
- Sentence structure issues: Line 16, "basic principle that " --> " basic principle of", Line 17, "while pushing negative pairs" --> "and pushing negative pairs".
- Inconsistent parallel structures: Line 20, "failure of" --> "the failure of", Line 21, "rigidness" --> "and the rigidness"

**Questions For Authors:**

What are the core technologies or solutions that play a critical role in modeling lightweight in this paper? Could these technologies or solutions be applied to other tasks?

**Relation To Broader Scientific Literature:**

The application of lightweight graph contrastive learning across multiple scientific domains demonstrates its versatility. For bioinformatics, this paper may integrate heterogeneous graph structures of multi-omics networks for gene-protein association prediction. For knowledge graphs, this paper can combine with knowledge-enhanced graph convolutional networks to enhance cross-entity reasoning accuracy.

**Theoretical Claims:**

I checked the correctness of the proof for Propositions 1-2 and Theorem 1.

---

> ### Author Rebuttal · Authors · 2025-03-30
>
> We sincerely appreciate the reviewer’s positive feedback and careful reading. Below, we will provide a point-by-point response.
>
> W1: See Q of **Reviewer m6R5**.
>
> **W2: This may be caused by software incompatibility. You can try Adobe Acrobat 10.0 to open the PDF file.**
>
> W3: We will modify this in the final version.
>
> C: We did consider using other types of graph encoders, such as Graph Transformers (GT). However, our decision to adopt GCN as the encoder was ultimately driven by the following factors:
>
> **1. Computational Efficiency and Model Lightweight**
>
> One goal of this work is to develop a lightweight graph contrastive learning framework. Compared with Graph Transformers, GCN offers lower computational complexity and fewer parameters, making it easier to train on large-scale graph data while maintaining robust performance.
>
> **2. Compatibility with Manifold Learning**
>
> Our method is rooted in manifold learning, emphasizing the preservation of local linear relations. GCN’s hierarchical neighborhood aggregation mechanism inherently aligns with this design philosophy, enabling the learned embeddings to better retain local geometric information. In contrast, Graph Transformers primarily rely on global attention, which, while effective for capturing long-range dependencies, may dilute the fidelity of local geometric structures.
>
> **3. Experimental Results and Comparability**
>
> To ensure fair comparisons with mainstream GCL methods, we adopted the same encoder setup. Since most baseline models use GCN, maintaining consistency ensures experimental comparability and highlights the advantages of our framework itself.
>
> Even so, we acknowledge the potential of Graph Transformers in modeling long-range dependencies. Future work could explore their integration into the ML²-GCL framework to further enhance global structural modeling capabilities.
>
> Q: The lightweight modeling in this paper primarily relies on the following core technologies and solutions, which reduce computational complexity while ensuring model effectiveness and generalization capability:
>
> **1. Manifold Learning-Driven Positive Pair Weight Calculation**
>
> We compute the positive pairs weight matrix W_p with locally linear embedding, enabling direct learning of global nonlinear representations on the original graph structure without pairwise distance computation.
>
> This strategy applies to other contrastive learning tasks requiring structural preservation, such as text and protein interaction networks. Additionally, it can be used for dimensionality reduction tasks, e.g., manifold-based image feature extraction.
>
> **2. Closed-Form Solution Optimization**
>
> We solve the positive pairs weight matrix W_p through a closed-form solution, which significantly reduces computational complexity compared with gradient descent-based optimization methods, making the approach more lightweight.
> This closed-form solution can be generalized to unsupervised dimensionality reduction, spectral clustering, and other tasks, particularly suitable for large-scale datasets such as graph neural network pretraining.
>
> **3. Single-View Modeling Avoiding Data Augmentation**
>
> Traditional graph contrastive learning relies on data augmentation to generate multiple views. This study implements contrastive learning directly on a single view, reducing storage requirements while eliminating semantic bias introduced by augmentation.
>
> This method applies to scenarios where data augmentation is difficult to define, such as social network analysis and biological graph applications, where complex data structures pose challenges in generating high-quality augmented views.
>
> The lightweight design in this study is not task-specific but based on rational improvements in optimization strategies, sampling methods, and modeling approaches. It can thus be generalized to other graph learning tasks, visual contrastive learning, and embedding learning in natural language processing. We believe these strategies can inspire the development of more efficient and scalable unsupervised learning methods.

---

> > ### Comment · Reviewer_bpNM · 2025-04-05
> >
> > Thank you for your response. As all of my concerns have been solved, I am glad to raise my score.

---

### Official Review · Reviewer_Ytzf · 2025-03-12

**Overall Recommendation:** 3

**Summary:**

Recent years have witnessed a phenomenon that graph contrastive learning faces the balance between effectiveness and efficiency. In spite of its popularity and success, several potential risks including underlying semantic disturbance brought by augmentation strategies, failure of GCN in capturing long-range dependence, rigidness and inefficiency of node sampling techniques  still remain to be solved. In this paper, authors provides a novel perspective for graph contrastive learning, called ML^2-GCL. It achieves global nonlinear structure recovery from locally linear fits, which can make up for the defects of GCN. The most amazing advantage is about the lightweight due to its closed-form solution of positive pairs weights and removal of pairwise distances calculation. A series of theoretical analysis and empirical performance completely demonstrate its superiority.

**Claims And Evidence:**

Yes, the claims are supported by clear and convincing evidence.

**Essential References Not Discussed:**

No, all the essential related works have been adequately discussed

**Experimental Designs Or Analyses:**

Yes

**Methods And Evaluation Criteria:**

Yes.

**Other Comments Or Suggestions:**

The authors should check some typos in Lines 16-17.

**Other Strengths And Weaknesses:**

1) Through introducing manifold geometric similarity metrics, such as local neighborhood preservation, the distributional consistency of positive and negative pairs in the latent space is improved.

2) Removal of both graph augmentation and pairwise distances calculation can reduce the computational costs significantly, which aligns with the current demand for efficient graph models.

3) Both theoretical analysis and empirical performance adequately verify the advantages in terms of effectiveness and lightweight.

4) I've noticed that the proposed novel contrastive loss function uses the closed-form solution of anchor node reconstruction combination weights. Authors had better give more detailed explanations.

5) In the Conclusion Section, future work is missing.

**Questions For Authors:**

What are the differences between personalized graph embedding and traditional graph embedding?

**Relation To Broader Scientific Literature:**

The cross-disciplinary innovation of lightweight graph contrastive learning with other technologies drives scientific literature development includes integration with traditional supervised learning and synergy with transfer learning. Specifically, in weakly supervised scenarios, contrastive loss compensates for the scarcity of annotated data. Cross-domain feature alignment enables knowledge transfer and model generalization.

**Theoretical Claims:**

Yes

---

> ### Author Rebuttal · Authors · 2025-03-30
>
> We sincerely appreciate the reviewer’s positive feedback and attention. Here, we will provide a point-by-point response.
>
> W1: Thank you for your attention. In fact, the proposed novel contrastive loss can be regarded as personalized graph embedding, where positive pairs with larger weights should be more similar, while negative pairs with smaller weights should be more different.
>
> W2: In future, we will investigate deep integration mechanisms between manifold learning and dynamic graph structures to address nonlinear evolution patterns in temporal data or dynamic networks. We can also extend such unsupervised contrastive paradigms to multimodal data for efficient computation in large-scale graph scenarios. The content will be supplemented upon acceptance of the paper.
>
> C:  We will modify "While existing works follow the basic principle that pulling positive pairs closer while pushing negative pairs far away..." to "While existing works follow the basic principle of pulling positive pairs closer and pushing negative pairs far away...".
>
> Q: Personalized graph embedding is a more generalized framework which assigns each positive pair or negative pair its own weight. Traditional graph embedding can be regarded as a special form of personalized graph embedding. In Appendix A.3, we have briefly discussed the differences.

---

### Official Review · Reviewer_m6R5 · 2025-03-12

**Overall Recommendation:** 4

**Summary:**

This paper explores an effective and lightweight graph contrastive learning method called ML^2-GCL, highlighting the need for a deeper understanding of graph contrastive learning methods from a manifold learning perspective. ML^2-GCL recovers global nonlinear structure from locally linear fits with closed-form solution of positive pairs weights, using which to design a novel contrastive loss function and update graph encoders. In addition, this paper proves the existence of the optimal closed-form solution and analyses the essence of ML^2-GCL. Extensive experiments verify both effectiveness and lightweight of ML^2-GCL.

**Claims And Evidence:**

Yes. This paper explores the marriage of manifold learning to graph contrastive learning for the first time, develops a new paradigm of effective and lightweight contrastive learning, and derives a series of theoretical analysis. These theoretical analysis proves the existence of the optimal closed-form solution of positive pairs reconstruction weights and reveals its connection with graph embedding, which provide strong evidence for the claim.

**Essential References Not Discussed:**

No. This paper does not omit any essential references.

**Experimental Designs Or Analyses:**

Yes. I checked the validity of the experimental designs and analyses, both of which are sound and complete.

**Methods And Evaluation Criteria:**

Yes. The proposed ML^2-GCL method is proper for the problem of GCL.

**Other Comments Or Suggestions:**

(a) Spelling errors: Line 31, " lightweightness " --> " lightweight".
(b)  The mathematical expressions in the framework diagram are partially truncated, affecting the readability of key equations.

**Other Strengths And Weaknesses:**

Strengths：
(a) This paper proposes a novel and insightful method for GCL. By incorporating manifold learning theory, i.e., the manifold smoothness assumption, ML^2-GCL optimizes the geometric constraints of graph node embeddings and enhances the capability of low-dimensional representations to characterize complex graph structures.
(b) By simplifying the contrastive pair generation process, ML^2-GCL reduces the computational complexity of traditional graph contrastive learning methods, especially those exploiting graph augmentation strategies.
Weaknesses
(a) The Conclusion Section lacks an exploration of potential future research directions.
(b) Publicly releasing the code and experimental details may ensure the reproducibility of the results and enhance the impact of this work.

**Questions For Authors:**

In general, graph encoding is the first step in GCL. However, in this paper, the computation process of positive pairs reconstruction weights does not include graph encoders, could authors explain why?

**Relation To Broader Scientific Literature:**

(a) Dynamic graph adaptation‌: Existing methods primarily focus on static graphs, necessitating exploration of temporal enhancement strategies‌.
‌(b) Interpretability improvement‌: Lightweight model simplification may compromise semantic interpretability, requiring optimization through causal inference and related methods‌.

**Theoretical Claims:**

Yes. I checked the correctness of the proofs for theoretical claims, which proves the existence of the optimal closed-form solution and the correlation between ML^2-GCL and graph embedding.

---

> ### Author Rebuttal · Authors · 2025-03-30
>
> Thanks for the reviewer’s careful reading and valuable suggestion. In the following, we will provide a point-by-point response.
>
> Wa: In future, we will investigate deep integration mechanisms between manifold learning and dynamic graph structures to address nonlinear evolution patterns in temporal data or dynamic networks. We can also extend such unsupervised contrastive paradigms to multimodal data for efficient computation in large-scale graph scenarios. The content will be supplemented upon acceptance of the paper.
>
> Wb: The code will be made publicly available upon acceptance of the paper. For experimental details, please refer to Appendix C.
>
> Ca: Thanks for the careful reading. We will modify this in the final version.
>
> **Cb: This may be caused by software incompatibility. You can try Adobe Acrobat 10.0 to open the PDF file.**
>
> Q: Thank you for the question. Our method, ML²-GCL, intentionally excludes the graph encoder when calculating the reconstruction weights for positive pairs. This is a deliberate design choice, primarily motivated by the following reasons:
>
> **1. Avoiding Encoder Interference to Ensure Geometric Consistency**
>
> Traditional GCL methods typically rely on embeddings from graph encoders to measure node similarity and construct positive/negative pairs. However, this approach is susceptible to instability caused by encoder initialization, training states, and parameter updates, which can lead to inconsistent positive pairs selection.
>
> In ML²-GCL, the calculation of positive pair weights is based on locally linear embedding-based geometric optimization, directly constructing the positive pairs weight matrix W_p in the raw feature space. This avoids biases introduced by graph encoders, ensuring that W_p is solely determined by the inherent geometric structure of the data, aligning with the principles of manifold learning.
>
> **2. Enhancing Robustness of Contrastive Learning**
>
> Computing W_p in the raw feature space can serve as prior information for the encoder’s input data, providing stable supervisory signals. In contrast, computing W_p after GCN processing would risk instability, as updates to GCN parameters could disrupt the construction of positive pairs. Furthermore, positive pair selection should depend on the graph structure and node features themselves, rather than the encoder’s initial state.
>
> **3. Reducing Computational Complexity and Improving Efficiency**
>
> Traditional GCL methods require similarity calculations in the encoder’s embedding space, whereas ML²-GCL computes W_p via a closed-form solution, significantly lowering computational costs and enabling a lightweight framework. Our method also avoids exhaustive pairwise similarity computations by focusing on local geometric structures, further enhancing efficiency.
>
> **4. Ensuring Generalizability Across Graph Encoders**
>
> Since W_p is encoder-independent, ML²-GCL can flexibly integrate with various graph neural networks, such as GCN, GAT, and GraphSAGE, without altering the positive pair construction strategy. This design ensures broad applicability, allowing ML²-GCL to adapt to diverse GNN architectures seamlessly.
>
> In summary, our design is grounded in manifold learning theory, ensuring stable positive pair construction, computational efficiency, and method generalizability. The experimental results validate the effectiveness of this approach, demonstrating its theoretical and practical soundness.

---

### Official Review · Reviewer_6NFU · 2025-03-13

**Overall Recommendation:** 3

**Summary:**

As a mainstream and representative unsupervised learning method, contrastive learning has achieved great success in the field of computer vision. Inspired by such achievements, graph contrastive learning (GCL) has attracted much interests in the past few years. Despite its excellent performance, GCL suffers from underlying semantic disturbance, rigid and inefficient node sampling, etc. To address this issue, this paper develop ML2-GCL, a Manifold Learning Inspired
Lightweight Graph Contrastive Learning method. It is the first exploration to marry manifold learning with graph contrastive learning, which avoids the semantic disturbance and high computational complexity completely. Theoretical analysis and experimental results demonstrates its effectiveness and lightweight.

**Claims And Evidence:**

Yes, they are.

**Essential References Not Discussed:**

As far as I know, no key reference is missing.

**Experimental Designs Or Analyses:**

Yes, I did.

**Methods And Evaluation Criteria:**

Yes, they do.

**Other Comments Or Suggestions:**

See Weaknesses.

**Other Strengths And Weaknesses:**

Strengths:

1. The idea is novel. This paper combines manifold learning with graph contrastive learning and proposes a lightweight design, which is in line with the current research trend of contrastive learning in reducing computational costs.

2. The technique is reasonable. It designs a novel contrastive loss function with the closed-form solution of anchor node reconstruction combination weights to achieve both effectiveness and lightweight.

3. Theoretical analysis proves the existence of the optimal closed-form solution. Extensive empirical studies on benchmark datasets demonstrate that the proposed method achieves state-of-the art performance in terms of effectiveness and lightweight.

Weaknesses:
1. To improve readability, authors are encouraged to discuss the similarities and differences between manifold learning and graph contrastive learning.

2. Authors should carefully check and revise a few spelling and grammar mistakes. For example, "While existing works follow the basic principle that pulling positive pairs closer while pushing negative pairs far away...", where " principle that pulling ... while pushing ..." should make some adjustments.

3. There are minor errors in Table 7. For the GPU memory usage on Amazon-Photo dataset, GRACE is the second least and AFGRL is the third least. The color is reversed.

**Questions For Authors:**

1. Are the positive pairs weight matrix Wp and negative pairs weight matrix Wn of equal size?

2. From Tables 5-6, I observe that the optimal parameter k is relatively small. Could authors give more explanation?

**Relation To Broader Scientific Literature:**

The application of lightweight graph contrastive learning in recommendation system has verified its universality. With lightweight representation of user product interaction graphs, it solves the problems of data sparsity and long tail recommendation.

**Theoretical Claims:**

Yes, I did.

---

> ### Author Rebuttal · Authors · 2025-03-30
>
> We sincerely appreciate the reviewer’s positive feedback and valuable comments. Below, we will provide a point-by-point response to each comment.
>
> W1:  We have briefly discussed this in Introduction. Here, we will give a detailed discussion.
>
> **Similarities**
>
> **1. Consistency in Core Objectives**
>
> Both aim to extract effective low-dimensional representations from complex data structures. Manifold learning reveals low-dimensional manifold structures embedded in high-dimensional data through dimensionality reduction, while graph contrastive learning learns discriminative embeddings through contrasting different graph structures or node relations.
>
> **2. Focus on Local Structures**
>
> Manifold learning algorithms preserve local geometric properties through constructing neighborhood relations. Graph contrastive learning enhances sensitivity to local structures through contrasting node or subgraph neighborhoods.
>
> **3. Advantages in Handling Nonlinear Relations**
>
> Manifold learning captures nonlinear low-dimensional structures of high-dimensional data. Graph contrastive learning excels at modeling non-Euclidean relations of graph data.
>
> **Differences**
>
> **1. Theoretical Foundations and Input Data Types**
>
> Manifold learning is based on topological manifold theory, assuming high-dimensional data lies on low-dimensional manifolds. Its inputs are high-dimensional vectors, such as images and text. Graph contrastive learning is rooted in graph theory and contrastive learning frameworks. Its inputs are graph-structured data, such as nodes and edges.
>
> **2. Methodological Approaches**
>
> Manifold learning reduces dimensionality via constructing local neighborhoods or global constraints, while graph contrastive learning generates positive/negative pairs and optimizes models using contrastive loss functions to distinguish similar/dissimilar structures.
>
> **3. Application Scenarios**
>
> Manifold learning is primarily used for data visualization, denoising and feature extraction. Graph contrastive learning is applied to graph classification, node classification and graph generation, particularly in scenarios with missing or improvable graph structures .
>
> **4. Mathematical Tools and Optimization Goals**
>
> Manifold learning relies on graph Laplacian matrices, geodesic distance calculations, and minimizes reconstruction errors or preserves local geometry. Graph contrastive learning uses contrastive loss to maximize similarity of positive pairs and minimize similarity of negative pairs, focusing on discriminability of representation spaces.
>
> W2: We will modify "While existing works follow the basic principle that pulling positive pairs closer while pushing negative pairs far away..." to "While existing works follow the basic principle of pulling positive pairs closer and pushing negative pairs far away...".
>
> W3: We will modify this in the final version.
>
> Q1: Yes, both of them are square matrices with dimension N.
>
> Q2: We observe that the optimal range of k values is generally small across all datasets, which can be primarily explained from the following two perspectives:
>
> **1. Theoretical Importance of Local Neighborhoods**
>
> In ML²-GCL, k determines the scope of the positive pair set P_i, where the positive samples of an anchor node are derived from its k-hop neighbors. Theoretically, a smaller k makes positive pairs more localized, enhancing the preservation of local geometric structures while reducing noise interference from distant neighbors. Our optimization objective, based on manifold learning, constructs the positive pairs weight matrix W_p through locally linear embedding, making it more suitable for information propagation within small-scale neighborhoods. When k becomes excessively large, distant positive pairs may introduce excessive noise, compromising the fidelity of manifold structures and resulting in overly homogeneous representations that degrade contrastive learning performance.
>
> **2. Empirical Observations on k-Sensitivity**
>
> Our experiments reveal that larger k values lead to performance degradation and reduced model discriminability. This may occur because a larger k introduces excessive positive pairs, potentially amplifying noise and destabilizing local structures. When k increases, more nodes are included in the positive pair set. However, information from many of these nodes may already have been propagated through GCN layers, leading to information redundancy and diminished model discriminative power.
>
> Based on the above analysis, we conclude that smaller k values allow ML²-GCL to better preserve manifold-localized structures while avoiding over-smoothing issues, thereby delivering superior experimental performance.

---

> > ### Comment · Reviewer_6NFU · 2025-04-06
> >
> > Thank you for the author's response. I have also reviewed the comments from the other reviewers and the corresponding replies from the author. I will maintain my score.

---

### Decision · Program_Chairs · 2025-05-01

**Decision:**

Accept (poster)

**Comment:**

This paper proposes Manifold Learning Inspired Lightweight Graph Contrastive Learning, which is well-structured and easy to understand. The reviewers generally acknowledge the paper's theoretical analysis and experimental validations. Several minor concerns have been also raised regarding grammatical issues, spelling errors and the lack of analysis on the neighborhood sampling method. I encourage the authors to address these comments in the final revision. I am confident that this paper can be accepted.